The role of circRNA in breast cancer drug resistance

Yang Shaofeng
Li Donghai 41031510@qq.com
Inner Mongolia Medical University Hospital , Hohhot , China
Uversky Vladimir
Electronic publication date: 2024 Dec 18
Publication date: 2024
Volume: 12
Electronic Location ID: e18733
Received 2024 Jun 20; Accepted 2024 Nov 27
Copyright: © 2024 Yang and Li
Copyright year: 2024
Copyright holder: Yang and Li
License: This is an open access article distributed under the terms of the Creative Commons Attribution License, which permits unrestricted use, distribution, reproduction and adaptation in any medium and for any purpose provided that it is properly attributed. For attribution, the original author(s), title, publication source (PeerJ) and either DOI or URL of the article must be cited.
License URL: https://creativecommons.org/licenses/by/4.0/

Keywords: BC, circRNA, Drug resistance, Antineoplastic

Funding: Natural Science Foundation of Inner Mongolia Autonomous Region 2022MS08010 Graduate Student Excellence Program YKDD2023ZY001 This article was supported by the Natural Science Foundation of Inner Mongolia Autonomous Region (Grant No. 2022MS08010) and Graduate Student Excellence Program (YKDD2023ZY001). The funders had no role in study design, data collection and analysis, decision to publish, or preparation of the manuscript.

==============================
Among women with cancer, breast cancer has surpassed lung cancer to become the most prevalent type of cancer globally. High-throughput sequencing of breast cancer tissues from many patients has revealed significant variations in circRNA expression across different types of breast cancer. Chemotherapy is currently a very important method for treating breast cancer; however, as the number of chemotherapy sessions increases and considering factors such as the patient’s immune response, drug resistance has become a challenging issue in treating breast cancer. It is well known that drug resistance is associated with multiple factors, and different resistance mechanisms involve different roles of circRNA. This review consolidates literature from the past 5 years and addresses the shortcomings in the broad description of circRNA’s role in breast cancer drug resistance. It categorizes and describes the drug resistance and its mechanisms in different types of breast cancer, as well as the roles of circRNA and signaling pathways in drug resistance.

Introduction

Breast cancer (BC) is the most common type of cancer among everyday diseases and is also one of the leading causes of cancer-related deaths in women (Britt, Cuzick & Phillips, 2020; Harbeck et al., 2019). Due to the heterogeneity among patients and the different molecular types of tumors, breast cancer has varied treatment options and prognoses (Lüönd, Tiede & Christofori, 2021). Different subtypes of breast cancer are related to hormone receptors observed in breast cancer cells (Perou et al., 2000). CircRNA expression levels vary by breast cancer classification. With the deepening of circRNA research, it has been found to have a close relationship with breast cancer. CircRNA is a novel non-coding RNA (ncRNA) formed through the back-splicing of protein-coding genes. It plays roles in various diseases and biological processes (Zhou et al., 2023). The internal structure of circRNA is a highly stable circular molecule, which protects it from degradation by nucleases, thus providing potential for cancer prognosis or diagnosis (Wu et al., 2023). In breast cancer, circRNA can act as a tumor suppressor, influencing tumor occurrence, development, apoptosis, cell cycle, and tumor microenvironment (Ghafouri-Fard et al., 2021). Particularly, the expression of exosomal circRNA is higher in breast cancer tissues (De Palma et al., 2022). This high expression phenomenon facilitates subsequent research. Additionally, circRNA is related to breast cancer drug resistance. There are various mechanisms of drug resistance in breast cancer, such as endocrine resistance mechanisms (resistance to tamoxifen or aromatase inhibitors), chemotherapy resistance mechanisms (drug efflux), DNA damage repair (DDR) mechanisms, tumor microenvironment (TME) and tumor cell crosstalk mechanisms, targeted therapy mechanisms, etc (Zeng et al., 2018). Currently, research on circRNA in relation to breast cancer drug resistance often presents a singular description. In this review, we summarize the above mechanisms and the impact of circRNA expression on the occurrence and development of breast cancer, providing an in-depth supplement to the role of circRNA in breast cancer treatment. We also emphasize the application of circRNA as a molecular marker in breast cancer therapy.

Circrna generation and regulation

CircRNA generation: CircRNA is formed in circular transcription by reverse splicing of early messenger RNA (mRNA). During eukaryotic transcription, there is always a competition between linear transcription and reverse splicing. The presence of long introns, RBPs (RNA-binding proteins), and inverted repeat motifs facilitates reverse splicing, and the downstream splicing donor is brought closer to the upstream splice acceptor site by base-pairing of RBPs or inverted repeat motifs (Kristensen et al., 2019). Recent results from Drosophila suggest that U2snRNP depletion increases circRNA production compared with its linear counterpart (Liang et al., 2017). So when mRNA production stops, the nascent RNA can be redirected to different pathways that promote reverse splicing to produce circRNAs (Liang et al., 2017). In addition to the defective splicing mechanism described above, the flanking intron sequences on either side of the exon (i.e., splice donor site and splice acceptor site) can support efficient cyclization of different exons across eukaryotes (Kramer et al., 2015). Dimerization of RBP can bind to specific sequences of these flanking introns and facilitate the reverse splicing process (Conn et al., 2015; Errichelli et al., 2017; Verheijen & Pasterkamp, 2017). Recent studies have shown that there exists an intron-containing circRNA called ElciRNA, and the nuclear localization of ElciRNA and its association with PolII suggests that it may be involved in the regulation of transcription U1 snRNAs and EIciRNAs bind to each other through binding sites in EIciRNAs, and all three of them begin to promote gene transcription in the presence of PolII (Li et al., 2015b) (Fig. 1).

Figure 1 circRNA production and regulation.

Circrna expression in breast cancer

Upregulation of circRNA in different types of breast cancer

The role of circRNA in TNBC has been confirmed. On one hand, CircRNA can act on a signaling pathway to promote the initiation of bone remodeling factors and stimulate breast cancer metastasis to bone (Zhang et al., 2021). On the other hand, circBCBM1 affects the miR-125a/BRD4 axis, promoting brain metastasis of breast cancer (Huang & Zhu, 2021). Additionally, circZEB1 promotes proliferation and reduces apoptosis through the miR-448/eEF2K signaling pathway (Pei et al., 2020). An interesting study shows that circSEPT9 is overexpressed in TNBC tissues, and its high expression level is associated with advanced clinical stages and poor prognosis. Mechanistically, circSEPT9 can regulate the expression of leukemia inhibitory factor (LIF) by sponging miR-637, thereby activating the LIF/Stat3 signaling pathway involved in TNBC progression (Yi et al., 2023). Types such as circZEB1, circSKA3, circRNA-UCK2, circPDCD11, circ-UBAP2, circGFRA1, circEPSTI1, and circHIF1A are also highly expressed in TNBC, and their high expression is significantly related to the prognosis and tumor progression of TNBC (Huang & Zhu, 2021; Lyu et al., 2021; Nielsen et al., 2022; Pei et al., 2020; Xiang et al., 2019; Xing et al., 2021; Zheng et al., 2021). It is noteworthy that high expression of circRNAs can not only promote breast cancer metastasis but also have inhibitory effects. CircCDYL is also overexpressed in TNBC, promoting apoptosis and inhibiting proliferation through the miR-190a-3p/TP53INP1 axis, thereby upregulating the tumor suppressor TP53INP1 protein in TNBC (Geng et al., 2020). In HER2-positive breast cancer, ISH analysis shows that circCDYL in tumor tissues is elevated nearly 1.63 times compared to adjacent HER+BC tissues, and HER2 gene does not guide circCDYL expression in HER2 cells (Liang et al., 2021). Previous studies have shown that upregulation of circCDYL enhances proliferation and autophagy in HER2 cells (Geng et al., 2020). However, the specific biological function of circCDYL in HER2+BC cells still requires further exploration. In this study, qRT-PCR experiments validated that specific siRNA can silence circCDYL, which can consistently decrease or increase circCDYL in HER2+BC cells; overexpression of circCDYL promotes activation of the PI3K/AKT pathway in SK-RB-3 cells. CCK8 assays indicated that AKT inhibitors can partially impair the overexpression of circCDYL, suggesting that circCDYL promotes HER2+ proliferation by activating the PI3K/AKT pathway (Liang et al., 2021). Furthermore, circRNA is also related to resistance to trastuzumab; for example, circ-BGN located in the cytoplasm of trastuzumab-sensitive and resistant cells, with knockdown of circ-BGN significantly restoring sensitivity to trastuzumab (Wang et al., 2020). Additionally, circRNA not only affects the tumorigenesis of ER-positive breast cancer but can also serve as a genetic regulator influencing resistance to endocrine therapy in ER-positive breast cancer (Treeck, Haerteis & Ortmann, 2023). For instance, circRNA_0025202 interacts with miR-182-5p to upregulate FOXO3a expression, inhibiting tumor growth and tamoxifen resistance (Sang et al., 2019). Another molecule, circPVT1, is highly expressed in breast cancer cells and promotes tumorigenesis and endocrine resistance through miRNA sponging and protein scaffolding effects (Yi et al., 2023).

Downregulation of circRNA expression in different types of breast cancer

CircRNA not only plays a role through upregulation in TNBC but also affects proliferation, migration, and apoptosis through downregulation. An earlier analysis indicated that circAHNAK1 is downregulated in TNBC and acts by sequestering miR-421. Since miR-421 can inhibit the expression of the tumor suppressor gene RASA1, reducing miR-421 serves to suppress TNBC proliferation and metastasis (Xiao et al., 2019). According to results shared by Wang et al. (2021), downregulation of circWAC can increase the sensitivity of TNBC to chemotherapeutic drugs. This mechanism involves the circWAC/miR-142/WWP1 regulatory loop affecting drug sensitivity (Dragomir & Calin, 2018). Additionally, there is a tumor suppressor-associated circRNA, circNR3C2, whose low expression in TNBC is associated with metastasis and mortality (Fan et al., 2021). It is well known that both estrogen and circRNA play roles in breast cancer development. However, it is still undetermined whether estrogen induces circRNA expression. To investigate this, Gao, Zhang & Zhao (2018) treated cultured MCF7 cells with ribonuclease digestion and RNase R treatment and performed RNA sequencing, also sequencing a control group. The results showed that the estrogen-induced group produced a large number of coding genes. Subsequent CIRI2 analysis of circRNA-seq (Gao, Zhang & Zhao, 2018) revealed 23,830 and 25,525 circRNAs in the control and estrogen-treated groups, respectively. Experiments confirmed that estrogen induces circRNA expression. Notably, circPGR was identified. Experimental findings indicated that knockout of circPGR can inhibit, while overexpression promotes cell cycle progression, colony formation, and cell migration. Furthermore, reduction of circPGR significantly diminishes estrogen-induced tumorigenesis. Thus, circPGR is essential for the growth and tumor development of ER-positive breast cancer cells (Wang et al., 2021).

New advances in circrna in breast cancer drug resistance

The issue of drug resistance in breast cancer treatment, including drugs such as doxorubicin (Al-Malky, Al Harthi & Osman, 2020), tamoxifen (Yao et al., 2020), paclitaxel (Gao et al., 2021), and even targeted therapies, is one of the main obstacles in breast cancer prevention and treatment, leading to higher mortality rates and poor prognosis (Lee, Tan & Oon, 2018; Vasan, Baselga & Hyman, 2019). Although multidrug resistance (MDR) is currently a major barrier in treating breast cancer, significant progress has been made in recent years regarding its molecular pathways and regulatory mechanisms. Among them, non-coding RNAs are one of the most important regulatory pathways in cells. circRNA, being one of the most important members of non-coding RNAs due to its unique structure of an endless circular sequence without a 3′ cap and 5′ poly(A) tail, is especially notable. This structure makes it more stable than other non-coding RNAs (Dragomir & Calin, 2018). Furthermore, circRNA participates in various signal pathway cross-talk (such as MAPK/ERK and PTEN/PIK3/AKT pathways), protein binding, as well as sponging microRNA and interfering with the splicing of other RNAs to alter multidrug resistance (Huang et al., 2020). This intertwines different types of breast cancer resistance, sharing a common research basis.

Drug efflux

Drug efflux is related to multidrug resistance and is a mechanism of cancer cell resistance (Chen et al., 2016). Breast cancer resistance protein (BCRP/ABCG) is a transmembrane transport protein and belongs to the ATP-binding cassette (ABC) transporter family. Because the substrates of this protein include anticancer agents, chemotherapy drugs and targeted therapies are often countered by this non-selective transporter protein (Hegedus et al., 2009; Hoque et al., 2020). Membrane-localized pumps play a functional role in the drug efflux mechanism, and BCRP is involved in the functioning of (Al-Malky, Al Harthi & Osman, 2020) these membrane-localized pumps (Emran et al., 2022). In cancer tissues, the gene encoding BCRP is upregulated, indicating that the expression of BCRP in cancer tissues increases, which limits the effective delivery of chemotherapy drugs. In this mechanism, BCRP actively transports intracellular drugs across the membrane in the opposite direction, pumping the drugs out of the tumor cells and reducing drug accumulation within the body (Gillet, Efferth & Remacle, 2007; Pesic et al., 2006). Moreover, BCRP also regulates and alters drug concentrations and distribution inside and outside the cells, mediating resistance through glutathione-dependent drug efflux and preventing the drugs from reaching their intended targets (Domenichini, Adamska & Falasca, 2019). Studies have found that certain circRNAs play significant roles in cancer resistance related to BCRP efflux transporters (Liu et al., 2021). In osteosarcoma, knocking out circ-CHI3L1.2 can downregulate the expression levels of enzymes and genes, thereby reducing drug resistance (Zhang et al., 2021). The ATP-binding cassette sub-family B member 1 (ABCB1) can reverse the effect of circRNA_103615 on drug resistance (Liang et al., 2021). This suggests that circRNA may also affect certain transmembrane transport proteins in breast cancer, and discovering such RNA’s role could be pivotal in overcoming breast cancer resistance in the future.

Autophagy, apoptosis, and cancer development

The growth of tumors essentially represents an imbalance between beneficial cells and tumor cells, with the proliferation rate of tumor cells exceeding their death rate forming the basis for cancer development (Morana, Wood & Gregory, 2022). Chemotherapy is a major current method for cancer treatment, and its effectiveness depends on the extent of cell damage and the ability to activate cell apoptosis (Sánchez-Carvajal et al., 2021). When tumor cell damage decreases and apoptosis ability is insufficient, cancer cell resistance occurs. Alternative splicing (AS) is an important mechanism in apoptosis. AS consists of four components: the spliceosome, splice sites, cis-acting elements, and trans-acting factors. The spliceosome recognizes the sites and catalyzes the removal of introns, while cis-acting elements and trans-acting factors bind to gene sites (Kumagai-Takei et al., 2018; Tilgner et al., 2012; Yang et al., 2014). Defects in AS frequently occur in cancer and may be related to gene mutations and AS mechanism disorders. In this process, circRNA inhibits AS by binding to miRNA, with circRNA 100/146 binding and collecting multiple miRNAs, promoting cancer invasion (Chen et al., 2019). In various cancers related to humans, tumor heterogeneity and treatment-induced resistance are attributed to different tumor stem cells, which can self-renew, evade cytotoxic cells, and increase the expression of resistance-related genes (Meacham & Morrison, 2013). Current research has found that stem cell autophagy has become a necessary condition for maintaining normal tissue stem cells, and like normal tissue stem cells, tumor stem cells also express autophagy. For example, in human ductal carcinoma in situ, autophagy becomes associated with invasive breast spheres (Espina et al., 2010), and can also sensitize estrogen-positive breast cancer to tamoxifen-induced cytotoxicity (Qadir et al., 2008). Autophagy has been shown to be involved in tumor cell survival and treatment resistance (Levy, Towers & Thorburn, 2017). In breast cancer, circDNMT1 is detected at high expression levels, where it binds with tp53 protein in the cytoplasm and enters the nucleus (Du et al., 2018). Excessive accumulation of tp53 in the nucleus promotes the expression of autophagy-related genes, increasing autophagy in breast cancer. Additionally, circDNMT1 also binds with HNRNPD/AUF1 and promotes its entry into the nucleus, upregulating DNMT1 expression. High DNMT1 expression then inhibits tp53 gene transcription (Du et al., 2018). Moreover, circCDYL can also promote autophagy. CircCDYL acts as a molecular sponge to adsorb MIR1275, leading to upregulation of ATG7 and ULK1 expression in cells, promoting autophagy and resulting in increased breast cancer cell proliferation (Ma et al., 2019).

circRNA and chemotherapy drug resistance

In breast cancer, the resistance to paclitaxel is closely related to circRNA. Research has found a high correlation between paclitaxel resistance in breast cancer and a substance called circular RNA angiogenic protein-like 1 (circAMOTL1). This substance participates in the regulation of the AKT pathway and promotes the expression of anti-apoptotic proteins, leading to paclitaxel resistance in breast cancer patients (Yang et al., 2020). Additionally, reports indicate that circ-ABCB10 binds with let-7a-5p, increasing paclitaxel sensitivity and related cell apoptosis, while also inhibiting the invasion and autophagy of paclitaxel-resistant breast cancer cells (Zang et al., 2020). Other studies have shown that the expression of circ-RNF111 is increased in paclitaxel-resistant tissues and cells, and knocking out circ-RNF111 reduces paclitaxel’s effectiveness on breast cancer cells (Liang et al., 2019). Further research could reveal that circRNA acts as a sponge for miRNA, and the enrichment of miRNA may affect paclitaxel resistance, suggesting that joint studies of circRNA and microRNA might yield significant findings regarding paclitaxel resistance. Tamoxifen, used as an endocrine therapy for advanced breast cancer, still faces the challenge of tamoxifen resistance. Studies have found that knocking out circBMPR2 promotes apoptosis and increases tamoxifen resistance in breast cancer cells, whereas overexpression of circBMPR2 reduces tamoxifen resistance in breast cancer (Hu et al., 2020). The mechanism involves circBMPR2 sponging miR-553, which upregulates the expression of ubiquitin-specific protease 4 (USP4), thus inhibiting breast cancer development and resistance. Additionally, a novel hsa_circ_0025202 has been discovered, which can inhibit tumors and sensitize tamoxifen both in vitro and in vivo through the miR-182-5p/FOXO3a axis. Furthermore, upregulation of circ_UBE2D2 also increases tamoxifen resistance in breast cancer cells, while downregulation has the opposite effect (Johnson-Arbor & Dubey, 2024). In ER-positive breast cancer cells with tumor-associated macrophages (TAM), circRNA UBE2D2 (circ-UBE2D2) is upregulated, and TAM-resistant cells can release exosomes containing circRNA UBE2D2 to induce drug-sensitive BC cells to develop TAM resistance. Mechanistically, circRNA UBE2D2 interacts with miR-200a-3p, subsequently reducing functional promotion of metastasis and increasing TAM resistance. Doxorubicin, one of the most effective chemotherapy drugs, still faces issues of survival and mortality due to breast cancer resistance. Doxorubicin resistance may be related to the state of molecular pathways in the apoptosis pathway and DNA double strands in target cells (Xie & Zheng, 2022). Most components of these molecular pathways are regulated by circRNA. Upregulation of Circ_0085495 leads to doxorubicin resistance in breast cancer. This mechanism sponges miR-873-5p, thereby enhancing doxorubicin resistance in breast cancer cells. Conversely, knocking out Circ_0085495 can inhibit tumor growth and increase the sensitivity of tumor cells to doxorubicin in vivo (Xie & Zheng, 2022). Besides Circ_0085495, circ_0006528 and Circ_0001667 also affect doxorubicin resistance (Cui et al., 2022; Zhou et al., 2018).

circRNA and breast cancer prognosis

According to current data, circRNA plays a very important role in chemoresistance of breast cancer. With advancements in medical research, circRNA has become a therapeutic target for overcoming breast cancer resistance (Gao et al., 2020). The dysregulation of circRNA is closely related to breast cancer resistance. CircRNA is involved in endocrine resistance of breast cancer; first, circRNA sequesters miRNA and forms a ceRNA network to exert its effects, and second, the upregulation of circ_UBE2D2 in tamoxifen-resistant BC cells suggests its potential role (Hu et al., 2020; Liang et al., 2019). Additionally, downregulation of circBMPR2 suppresses tamoxifen-induced apoptosis, and restoration can reverse resistance (Nedeljković & Damjanović, 2019). Therefore, targeting circRNA-miRNA networks might be an effective approach for treating endocrine resistance in BC patients. Another factor is the impact of signaling pathways. Increasing evidence points to the relationship between breast cancer resistance and signaling pathways, including the loss of PTEN (phosphatase and tensin homolog) and the activation of PAM (phosphatidylinositol 3-kinase (PI3K)/protein kinase B (AKT)/mammalian target of rapamycin (mTOR) signaling pathways), which significantly affect breast cancer resistance (Ali et al., 2017). Targeting signaling pathways to regulate downstream gene expression might be a method to eliminate BC resistance. Third, there is evidence that DNA damage repair (DDR) mechanisms are also involved in the development of breast cancer resistance. DDR mechanisms participate in repairing DNA damage and maintaining genome integrity and stability. CircRNA acts through these pathways in breast cancer prognosis; for example, elevated levels of circCDYL in breast cancer serum and tissues are associated with shortened survival and circCDYL can be used as a prognostic marker for breast cancer (Li et al., 2023). Furthermore, Hsa_circ_0067842 is elevated in breast cancer and is associated with worse outcomes (Smid et al., 2019). Plasma levels of hsa_circ_0001785 are closely related to breast cancer histological grading, TNM staging, and distant metastasis, indicating that hsa_circ_0001785 in plasma has potential diagnostic and predictive capabilities (Liu et al., 2018).

circRNA and breast cancer diagnosis

The uniqueness and biological functions of circRNA indicate its role in tumor growth, replication, metastasis, invasion, and drug resistance, suggesting that circRNA can be used as a biomarker for tumor diagnosis (Alhasan et al., 2016). Because circRNA can evade degradation induced by nucleases and, due to its circular structure, is more soluble in blood and plasma compared to linear RNA (Li et al., 2015a). Due to its stability and tissue specificity, circRNA has already been used as a biomarker for gastric cancer and liver cancer (Kulcheski, Christoff & Margis, 2016). With advancements in high-throughput sequencing and bioinformatics technologies, there is evidence suggesting that circRNA can contribute to tumorigenesis, providing new methods for identifying diagnostic biomarkers (Jiang et al., 2023).

circRNA and breast cancer treatment

Breast cancer stem cells (BCSCs) can resist DNA damage induced by chemotherapy drugs, leading to drug resistance (Mehraj et al., 2021). Interfering with DNA to induce mutations may increase the effectiveness of chemotherapy drugs. Tumor microenvironment (TME) cell crosstalk also promotes BC resistance. Various cells in the microenvironment interact, causing genetic alterations and uncontrolled growth and poor development, leading to hypoxia and acidosis in the microenvironment, which reshapes the TME. This reshaping of the TME can, in turn, affect cells by activating their resistance genes (Memczak et al., 2013). Additionally, circRNA can serve as a potential therapeutic target for breast cancer. The development of breast cancer involves multiple factors, and targeting downstream genes or proteins with circRNA is an important factor. circRNA RHOT1 promotes breast cancer progression by targeting the miR-204-5p/PRMT5 axis (Qu et al., 2019). The occurrence and progression of breast cancer are associated with dysregulated circRNA, so precisely targeting upregulated circRNA for unique back-splicing junctions (BSJ) for RNA interference could have therapeutic effects against tumors. Conversely, downregulating circRNA concentrations may also have specific anticancer effects. From a pharmacokinetic perspective, the high stability and long half-life of circRNA determine this effect (Sang et al., 2018). There have been exemplary studies on circRNA for breast cancer treatment. For instance, Sang et al. (2018) demonstrated that systemic injection of specific siRNAs targeting ciRS-7 inhibited migration and invasion of TNBC cells in a TNBC PDX mouse model, indicating that oncogenic ciRS-7 might be a potential therapeutic target for TNBC. They also reported that knockdown of ciRS-7 inhibits liver and lung metastases of TNBC cells in vivo by acting as a ceRNA in the sponge miR-1299. Thus, ciRS-7 exerts tumor-suppressive effects by regulating members of matrix metalloproteinases. Therefore, targeting the ciRS-7/miR-1299/MMP axis may be a new therapeutic strategy for TNBC (Zheng et al., 2021). Another strategy is that circRNA can sponge miRNA and RBP, becoming a potential carrier for breast cancer therapy. Zheng et al. (2021) validated that circGFRA1, as a sponge for miR-361-5p, promotes TNBC resistance to PTX through the TLR4 pathway. In vitro, silencing circGFRA1 upregulated the expression of miR-361-5p in PTX-treated TNBC PDX mouse models. Western blot results showed that after injecting lentivirus (LV-si-circGFRA1), TLR4 expression in PTX-treated mice was reduced. Furthermore, tumor size also decreased. Conversely, circGFRA1 and TLR4 were highly expressed in PDX mice treated with PTX alone, and tumor size increased (Tanaka et al., 2019). Additionally, circRNA can also regulate the expression of immune inhibitors through ceRNA networks, promoting tumor escape from immune surveillance. Tanaka et al. (2019) demonstrated that targeting CDR1-AS increases PD-L1 levels in CRC cells, induces T cell apoptosis, inhibits T cell activation and proliferation, ultimately leading to cancer immune escape (Conn et al., 2015). These methods provide new insights into breast cancer treatment, and more new mechanisms in circRNA therapy for breast cancer are yet to be discovered.

Discussion

Different subtypes of breast cancer have different circRNAs with high expression. Specifically, these high-expression circRNAs are related to two forms: one is the interaction and enrichment among miRNA-circRNA-lncRNA, which contributes to breast cancer drug resistance; the other is the formation of multiple drug resistances through different signaling pathways. While the processes may differ, the outcome converges on multiple drug resistance targets. We can step back and consider what affects the upstream expression of circRNAs to achieve multiple drug resistance in downstream target genes. For non-coding RNAs or signaling pathways, there might be some common factors stimulating circRNAs from upstream to control downstream targets. However, it is undeniable that circRNAs, as a highly conserved type of non-coding RNA, play a significant role in downstream enrichment or upstream integration. We could potentially produce point-to-point corrections targeting specific circRNAs, aiming to alter the adverse direction of breast cancer and possibly overcome current multiple drug resistance. It is worth noting that circRNAs, while regulating downstream target genes, can also serve as a cancer treatment method, participating in tumor occurrence and development. In the fight against breast cancer, circRNAs are playing an increasingly important role, and in-depth research on circRNAs could lead to essential breakthroughs in breast cancer treatment.

Additional Information and Declarations

Competing Interests

Author Contributions

Data Availability

The authors declare that they have no competing interests.

Shaofeng Yang conceived and designed the experiments, performed the experiments, analyzed the data, prepared figures and/or tables, authored or reviewed drafts of the article, and approved the final draft.

Donghai Li conceived and designed the experiments, performed the experiments, analyzed the data, prepared figures and/or tables, authored or reviewed drafts of the article, and approved the final draft.

The following information was supplied regarding data availability:

This is a literature review.

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
