# Peer review of "The role of circRNA in breast cancer drug resistance"

_PeerJ, doi:10.7717/peerj.18733_

## Round 0.1 · original submission · Major Revisions

Please address concerns of both reviewers and amend the manuscript accordingly.

Reviewer 1 ·

Basic reporting

The topic is relevant and significant; however, both the introduction and abstract section require rewriting.

Experimental design

The organization of the content could be enhanced for better clarity and coherence.

Validity of the findings

No comment.

Additional comments

The manuscript entitled “The role of circRNA in breast cancer drug resistance” by Shaofeng et al. reviews the functions and mechanisms of circRNA and explores their role in the treatment of drug-resistant breast cancer. However, I believe the manuscript, in its current form, cannot be considered appropriate for publication , and I have listed several points to improve the quality of the paper.

1. The abstract offers a brief overview of the article's purpose; however, it lacks clarity regarding the objectives, resulting in a disjointed presentation of the main elements. The authors are encouraged to revise the abstract to articulate their aims more precisely.

2. The keywords are inappropriately presented; it is recommended that the authors use the MeSH database to ensure the inclusion of relevant keywords.

3. The introduction does not adequately prepare the reader to engage with the discussion on the role of circular RNAs in drug resistance related to breast cancer. It would be beneficial to enhance this section by clearly articulating the significance of the topic and distinguishing it from similar subjects.

4. Numerous writing and grammatical errors are present throughout the manuscript. It is essential that sentences begin with a capital letter. Additionally, the spacing between words and references should be corrected. The purpose of the square brackets found in lines 196 and 202 requires clarification.

5. Section two (circRNA generation and regulation) should be expanded to provide a more comprehensive discussion on the role of circular RNAs.
6. It is advisable to restructure the third (circRNA expression in breast cancer) section by categorizing it into subheadings that reflect either increasing or decreasing expressions of circular RNAs (e.g., CircRNA and TNBC, decreasing expression of circular RNAs, increasing expression of circular RNAs, etc.).
7. The organization of content in the fourth section is commendable; however, subheadings 4.1, 4.2, and 4.3 do not align well with sections 4.4 to 4.6. To enhance coherence and maintain the flow of the article, the authors should standardize the content of this section, potentially classifying it by cellular mechanisms or drug types.

8. The details presented in the fourth section should be elaborated upon, and a thorough literature review is necessary to include the most current research in this area. Additionally, it is advisable for the authors to utilize up-to-date articles from databases to ensure the relevance of the cited literature.

9. While the fifth section provides valuable information, it is recommended that the authors divide this section into subsections addressing prognosis, diagnosis, and treatment.

10. Incorporating a graphic diagram that illustrates the synthesis of circular RNAs and their role in drug resistance in breast cancer would significantly enhance the scientific quality of the article.

Cite this review as

Reviewer 2 ·

Basic reporting

no comment

Experimental design

no comment

Validity of the findings

no comment

Additional comments

Yang and Li were intended to review the function of a novel RNA molecule called circRNAs in breast cancer drug resistance. They started by introduction circRNA biogenesis and regulation, circRNA expression in three subtype of breast cancer and summarized circRNA function in drug resistance. Although collected a series of publications, the review is not clear enough to help reader understand the related field, due to lacking summarized tables and figures, also with unprofessional statements and wording as listed below:

1. At least some figures or tables need to be added including breast cancer subtype, circRNA biogenesis, circRNA functions, circRNA functions in breast cancer drug resistance.
2. Missed a section of method for circRNA targeting to show the potential therapeutic method is feasible.
3. circRNA is formed by “back splicing” process not “reverse splicing”, the author also called the BSJ site (back splicing junction site).
4. “the next 5' splice site is linked to the 3' splice site of the previous mRNA”. We called “the downstream 3’ splicing site” and “the upstream 5’ splicing site”. CircRNAs are derived from “pre-mRNA” not “previous mRNA”.
5. “Unlike circRNAs produced by incomplete splicing”. What is “incomplete splicing”? How could this generate circRNAs?
6. “Although circRNAs have a linear 5'-3' structure”. There’s no linear structure but circular structure with shared sequence component with its linear host gene.
7. Line 77-80. I didn’t see the logic. RBP and sequence in the flanking intron could facilitate circRNA biogenesis. Why it is because there’s circRNAs contains both intron and exons?
8. CircRNA naming system is such a mess. There’s circSKA3, circRNA-UCK2, circ-UBAP2, circRNA UBE2D2, circRNA_0025202. Not to mention there’s “circIKBKBKB” which I don’t know if this is a typo or there’s a circRNA called this.
9. Considering no data is generated by review paper, I’m confusing with these statements: Line 110-113: In this study, we detected by qRT-PCR…
Line 122-125: To check whether estrogen induces the expression of circRNA, we subjected the cultured…
10. Clearly mistake in Line147-149: loop sequence formed by the 3' plus cap and 5' poly A tail. CircRNA have the loop structure, why “do not have”? Is circRNA highly conserve? No reference cited and to my knowledge, it is not highly conserved as other non-coding RNA does. Have a loop structure is not a reason to claim circRNA is “particular important”.

Cite this review as

---

## Round 0.2 · Major Revisions

Please carefully address all the critiques of the reviewer and make all the required changes in the manuscript.

Reviewer 1 ·

Basic reporting

The topic is both relevant and significant; however, it requires further review and revision to enhance its clarity and impact.

Experimental design

The revised version shows improvements compared to the original.

Validity of the findings

No comment.

Additional comments

The revised manuscript demonstrates some improvements; however, further refinement is necessary for publication. The manuscript should be drastically revised in line with the specific comments described below before reconsideration. Authors should enthusiastically and faithfully respond to each comment for improvement in the revised version.
1. The keywords require correction, particularly ensuring that the reference to the Brest concert is precise.
2. The phrase "Various levels of circular RNA (circRNA) expression are associated with different subtypes" lacks clarity in the introduction and should be rephrased for better understanding.
3. Although "exosomal circRNA" is mentioned in the introduction, its significance is not addressed in the manuscript. It would be beneficial to include a dedicated section discussing its relevance.
4. Regrettably, the authors did not adhere to the reviewer guidelines in the modified version and instead attempted to provide new responses by merely rephrasing previous sentences. For instance, Section 2 requires substantial revision as previously indicated. Additionally, the scientific inaccuracies present in the new version raise concerns. The statement "On one hand, IKBKB activates the NF-κB pathway, inhibiting IκBα feedback and promoting" should reference a circRNA that has been omitted in the latest revision. The authors must thoroughly verify all data presented, adhere to established writing conventions, and revise the manuscript in accordance with the feedback provided.
5. The modified manuscript does not include any figures, which is a notable omission.

Cite this review as

---

## Round 0.3 · Major Revisions

As you can see, the reviewer still thinks that additional work is required. Please add a graphical abstract figure and revise section 2 as requested by the reviewer.

Reviewer 1 ·

Basic reporting

The topic is both relevant and significant; however, it requires further review and revision to enhance its impact.

Experimental design

No comment.

Validity of the findings

No comment.

Additional comments

One of my suggestions for enhancing the scientific quality of the article was to include a graphic abstract. Unfortunately, the authors did not address this in the first revision and responded in the second revision with: "This is a review article and does not involve data processing, thank you."!!!!

Additionally, Section 2 requires major revision, and there are concerns regarding scientific accuracy. The statement "On the one hand, IKBKB activates the NF-κB pathway, inhibits the feedback of IκBα, and causes promotion" in lines 93-95 needs correction.

Cite this review as

---

## Round 0.4 · accepted · Accept

All issues pointed by the reviewers were addressed and revised manuscript is acceptable now.

Reviewer 1 ·

Basic reporting

No comment.

Experimental design

No comment.

Validity of the findings

No comment.

Additional comments

No comment.

Cite this review as